# NMR Structure of the FIV gp36 C-terminal Heptad Repeat and Membrane-Proximal External Region

**DOI:** 10.3390/ijms21062037

**Published:** 2020-03-16

**Authors:** Manuela Grimaldi, Michela Buonocore, Mario Scrima, Ilaria Stillitano, Gerardino D’Errico, Angelo Santoro, Giuseppina Amodio, Daniela Eletto, Antonio Gloria, Teresa Russo, Ornella Moltedo, Paolo Remondelli, Alessandra Tosco, Hans L. J Wienk, Anna Maria D’Ursi

**Affiliations:** 1Department of Pharmacy, University of Salerno, Via Giovanni Paolo II, 134, I-84084 Fisciano, Italy; magrimaldi@unisa.it (M.G.); mbuonocore@unisa.it (M.B.); mscrima@unisa.it (M.S.); istillitano@unisa.it (I.S.); asantoro@unisa.it (A.S.); daeletto@unisa.it (D.E.); moltedo@unisa.it (O.M.); tosco@unisa.it (A.T.); 2Institute of Polymers, Composites and Biomaterials, National Research Council of Italy, V.le J.F. Kennedy 54 - Pad. 20 Mostra d’Oltremare, 80125 Naples, Italy; angloria@unina.it (A.G.); teresa.russo@unina.it (T.R.); 3Department of Chemical Sciences, University of Naples Federico II, Monte S. Angelo, Via Cinthia, 80126 Naples, Italy; gerardino.derrico@unina.it; 4Department of Medicine, Surgery and Dentistry “Scuola Medica Salernitana”, University of Salerno, Via S. Allende, 84081 Baronissi (SA), Italy; gamodio@unisa.it (G.A.); premondelli@unisa.it (P.R.); 5Department of Chemistry, NMR Spectroscopy, Bijvoet Center for Biomolecular Research, Utrecht University, Padualaan 8, 3584 CH Uthrecht, The Netherlands; h.l.j.wienk@uu.nl; 6Division of Biochemistry, The Netherlands Cancer Institute, Plesmanlaan 121, 1066 CX Amsterdam, The Netherlands

**Keywords:** HIV, FIV, MPER, envelope glicoproteins, NMR

## Abstract

Feline immunodeficiency virus (FIV), a lentivirus causing an immunodeficiency syndrome in cats, represents a relevant model of pre-screening therapies for human immunodeficiency virus (HIV). The envelope glycoproteins gp36 in FIV and gp41 in HIV mediate the fusion of the virus with the host cell membrane. They have a common structural framework in the C-terminal region that includes a Trp-rich membrane-proximal external region (MPER) and a C-terminal heptad repeat (CHR). MPER is essential for the correct positioning of gp36 on the lipid membrane, whereas CHR is essential for the stabilization of the low-energy six-helical bundle (6HB) that is necessary for the fusion of the virus envelope with the cell membrane. Conformational data for gp36 are missing, and several aspects of the MPER structure of different lentiviruses are still debated. In the present work, we report the structural investigation of a gp36 construct that includes the MPER and part of the CHR domain (^737-786^gp36 CHR–MPER). Using 2D and 3D homo and heteronuclear NMR spectra on ^15^N and ^13^C double-labelled samples, we solved the NMR structure in micelles composed of dodecyl phosphocholine (DPC) and sodium dodecyl sulfate (SDS) 90/10 M: M. The structure of ^737-786^gp36 CHR–MPER is characterized by a helix–turn–helix motif, with a regular α-helix and a moderately flexible 3_10_ helix, characterizing the CHR and the MPER domains, respectively. The two helices are linked by a flexible loop regulating their orientation at a ~43° angle. We investigated the positioning of ^737-786^gp36 CHR–MPER on the lipid membrane using spin label-enhanced NMR and ESR spectroscopies. On a different scale, using confocal microscopy imaging, we studied the effect of ^737-786^gp36 CHR–MPER on 1,2-dioleoyl-sn-glycero-3-phosphocholine/1,2-dioleoyl-sn-glycero-3-phospho-(1’-rac-glycerol) (DOPC/DOPG) multilamellar vesicles (MLVs). This effect results in membrane budding and tubulation that is reminiscent of a membrane-plasticizing role that is typical of MPER domains during the event in which the virus envelope merges with the host cell membrane.

## 1. Introduction

Feline immunodeficiency virus (FIV) is a lentivirus infecting T lymphocytes in cats and causing an immunodeficiency syndrome, which is very similar to the one caused by HIV [1,2,3,4] in humans. The study of FIV and the design of molecules capable of preventing FIV infection in cats poses an important impulse on veterinary medicine to develop an effective FIV vaccine or therapy [5]. Furthermore, FIV represents a relevant model of pre-screening studies for HIV.

Lentiviruses enter host cells due to specific envelope glycoproteins that boost the interaction of the virus with the receptors on the host cell surface. Gp36 in FIV and gp41 in HIV catalyse virus envelope fusion with the respective host cell membranes. They show a common structural framework (Figure 1) [6,7,8] that includes the fusion peptide (FP), N-terminal heptad repeat (NHR), C-terminal heptad repeat (CHR) and membrane-proximal extracellular region (MPER). MPER is a hydrophobic, Trp-rich region (Figure 1) characterized by a strong membrane affinity. Virus entry is successful if NHR and CHR fold back to form a low-energy stable six-helical bundle (6HB) [9]; during this process, MPER plays an active role by correctly driving the virus membrane to assemble with the host cell membrane.

The design of molecules interfering with the formation of the 6HB or inhibiting the correct positioning of MPER on the membrane is a strategy currently being used to design new antiviral compounds acting as inhibitors of virus entry.

Peptides corresponding to fragments of gp41 or gp36 have been exploited as antiviral drugs or vaccines [10,11,12]; given the immunogenic potential of gp41 MPER, peptides from this region have been considered to develop anti-HIV vaccines [13,14,15,16], while peptides derived from gp36 MPER have been considered [6,15,17] for their ability to reduce FIV infectivity [18]. In this context, the 20-mer gp36 L^767^-M^786^, the octapeptide gp36 W^770^-I^777^ (C8), and the hexapeptide D^772^-I^777^(C6a) were previously identified as molecules exerting antiviral activity by preventing the penetration of FIV into the host cells. These peptides, endowed with a strong membrane binding property, were studied using several physico-chemical techniques; important information regarding their conformational properties was acquired using NMR-based conformational analysis [15,19,20]. Extending our previous work, using DNA recombinant techniques, we designed and produced a small protein corresponding to the gp36 fragment L^737^-M^786^. This polypeptide, including the 20- and 8-mer fragments previously analysed by us, encompasses the entire gp36 MPER and part of its adjacent CHR region. In the present work, we report the NMR analysis of ^15^N/^13^C-enriched ^737-786^gp36 CHR–MPER in a dodecyl phosphocholine/sodium dodecyl sulfate (DPC/SDS) micelle solution. Then, we investigated the positioning of ^737-786^gp36 CHR–MPER on the micelle surface and the 1,2-dioleoyl-sn-glycero-3-phosphocholine/1,2-dioleoyl-sn-glycero-3-phospho-(1’-rac-glycerol) (DOPC/DOPG) phospholipid bilayer using spin label-enhanced NMR experiments and ESR spectroscopy. On a larger scale, the effect of ^737-786^gp36 CHR–MPER on the size and shape of DOPC/DOPG multilamellar vesicles was studied by confocal microscopy imaging.

Despite the biological importance of MPER regions for the entry of lentiviruses into the respective host cells, several issues regarding their structure and mode of action are still obscure. Moreover, while MPER, together with several other fragments of gp41, have been studied by NMR and X-ray crystallography under a wide range of conditions [21,22,23,24], the structure of gp36 is almost unexplored. Only recently, data based on mass spectrometry confirmed the tendency of gp36 to form a six-helical bundle, which is analogous to gp41 [15,19,20,25,26,27,28,29,30,31,32,33,34].

In this context, our data provide the first high-resolution structure of gp36 MPER linked to CHR. They represent a useful contribution for clarifying the MPER molecular mechanism and its role in the interactions with the host cell membranes, with adjacent domains and the other gp36 monomers for the formation of the trimer.

The structure of ^737-786^gp36 CHR–MPER is discussed and compared with the structure of the corresponding region of gp41 and gp36 peptides that we previously studied. The specific disposition of Trp sidechains represents a key structural element, enabling a common molecular mechanism even in the presence of different amino acid sequences [35]. These elements inspire the design of new potential entry inhibitors that interfere with the formation of the trimer or replace MPER in the interaction with the lipid membrane.

## 2. Results

### 2.1. Circular Dichroism of ^737-786^gp36 CHR–MPER

Circular dichroism (CD) spectra of ^737-786^gp36 CHR–MPER were recorded in mixed DPC/SDS (90:10 M/M) micelles and in MLVs composed of DOPC/DOPG 90:10 M/M vesicles [36,37,38,39]. Both MLVs and micelle solutions are often used as biomimetic membrane models to study the structural features of membranotropic molecules. Micelles are made of surfactants, including zwitterionic DPC and negatively charged SDS, at much higher concentrations than their critical micelle concentration (c.m.c.). These surfactants form spherical aggregates, in which the polar headgroups are located on the surface, and the hydrophobic tails point to the centre. Micelle solutions are ideal systems for solution CD and NMR spectroscopy experiments, as they tumble rather quickly, resulting in high-resolution spectral lines [34,40,41].

Figure 2 shows the CD spectra of ^737-786^gp36 CHR–MPER collected in DPC/SDS 90:10 M/M and DOPC/DOPG 90:10 M/M MLVs. A qualitative analysis of CD curves indicates that ^737-786^gp36 CHR–MPER has similar conformational propensities in the two solvent systems. Quantitative estimation of CD data using the DICHROWEB website with CONTIN and SELCON3 analytical algorithms [42] shows that ^737-786^gp36 CHR–MPER in the DPC/SDS membrane mimicking system assumes 58% α-helix, 9% β-sheet and 33% random coil conformations, while in DOPC/DOPG, it assumes 69% α-helix, 4% β-sheet and 27% random coil conformations.

### 2.2. NMR Structure Determination

We expressed and purified a recombinant protein that includes the CHR and MPER segments of gp36. The structure and backbone dynamics of ^737-786^gp36 CHR–MPER were studied using solution NMR spectroscopy in mixed DPC/SDS (90:10 M/M) micelles [34,40,43]. Two-dimensional (2D) ^1^H-^13^C-HSQC/^1^H-^15^N-HSQC spectra of (^15^N-^13^C)-labelled ^737-786^gp36 CHR–MPER show all the characteristics of a stably folded protein, with 48 well-dispersed amide chemical shifts and uniform resonance line widths (Appendix A). Backbone chemical shift assignment was carried out by iteratively analysing 2D ^1^H-^13^C-HSQC/^1^H-^15^N-HSQC spectra and 3D HNCO, and HN(CA)CO spectra (SPARKY software) [44]. A similar procedure was used for chemical shift assignments of protein side chains by inspecting HBHA(CO)HN, CBCA(CO)HN, and HNCACB heteronuclear spectra [45,46,47]. ^1^H, ^13^C, and ^15^N chemical shifts are reported in the Appendix A.

The secondary chemical shifts of ^13^Cα, which refer to the difference between the observed chemical shifts and the corresponding residue-specific random coil values, are sensitive indicators of the local secondary structure [48,49]. Large positive values of ^13^Cα secondary chemical shifts suggesting α-helix structure are observed for the residues T^739-^E^748^ and L^751-^E^754^, which correspond to the CHR region. Conversely, the ^13^Cα chemical shift values of Q^755-^W^770^ and I^777-^M^786^ are typical of random coil structures, indicating that these regions are dynamically disordered in solution (Figure 3).

The structure of ^737-786^gp36 CHR–MPER in the micelle-bound state was derived by simulated annealing calculations, based on 443 sequential and short-range NOE distance restraints and 72 backbone dihedral angle restraints (Appendix A). The prediction of backbone dihedral angles was provided by the TALOS+ program [50]. Regular sequential and medium-range NOE effects, N,N(i, i+2), α,N(i, i+2), α,N(i, i+3) and α,β(i, i+3) (Figure 3), were observed in three-dimensional nuclear Overhauser enhancement (3D-NOESY) spectra collected in DPC/SDS micelle solutions. These were translated into interprotonic distances using the CALIBA routine of CYANA 2.1 software [51] and were used as restraints in NMR structure calculations (Appendix A). Statistics for the final NMR ensemble are summarized in Table 1 [52].

Figure 4 shows the NMR structure bundle of ^737-786^gp36 CHR–MPER as derived from the CYANA and TALOS+ calculations [50]. The structures show a good level of structural definition by the residues E^748^-G^760^ and Q^765^-V^774^, resulting in a 1.10 Å root-mean-square deviation (RMSD) value for the backbone. Quantitative estimation of backbone dihedral angles, according to PROMOTIF [53], using Kabsh and Sanders parameters, points to the presence of the following secondary structure segments: (i) α-helix for T^739^-Y^747^ residues, (ii) 3_10_ helix for E^748^-G^760^ and Q^765^-L^784^ residues, (iii) type II β-turn for K^761^-I^764^ residues. This last segment characterized by a higher flexibility defines a kink segment regulating the reciprocal position of the T^739^-Y^747^ α-helix and E^748^-G^760^ 3_10_ helix, with an average angle of 43.9° (Figure 4A). The structure was deposited in the Protein Data Bank (PDB ID: 6FTK). The flexibility of the interhelical linker suggests that the relative orientation and position on the lipid surface are subject to large dynamic disorder, and the structure shown in Figure 4A represents an average view.

The dynamics of the ^737-786^gp36 CHR–MPER backbone were determined by measuring relaxation parameters by solution NMR. Figure 5 reports the values of the T_1_, T_2_, and steady-state NOE values of the amide resonances plotted against the residue numbers. The NOE, T_1_, and T_2_ values are correlated with the mobility of each amide in the protein backbone, and this affects the rate at which an excited nucleus can explore the fluctuating fields around it to exchange energy and relax [54]. The relaxation data point to three different motional regimes: a rigid N-terminal portion (residues 738-757) with NOE values of approximately 0.6 and 0.8, long values of T_1_ and shorter values of T_2_; a flexible loop (residues 758-763) with faster motions and heteronuclear NOE values as low as 0.4; and a moderately flexible C-terminal region (residues 764-783), which includes residues with fast internal motion (Trp770 and Val774). These are characterized by the longest T_2_ values and particularly low NOE values. Figure 5 reveals that the N- and C-terminal helices have very similar T_1_/T_2_ ratios; this may be consistent with the two helices oriented at a 43.9° angle, resulting from the CYANA-simulated annealing. The experimental data are enforced by the random coil index (RCI) obtained by plotting the S_2_ values predicted by TALOS+.

### 2.3. NMR Spin-Label Analysis

NMR analysis in DPC/SDS 90:10 M/M micelle solution was the starting point to investigate the positioning of ^737-786^gp36 CHR–MPER on the micelle aggregates. For this purpose, we recorded NMR spectra of ^737-786^gp36 CHR–MPER in the presence of 5-doxylstearic acid (5-DSA), which was included in the micelles as paramagnetic probes. 5-DSA contains a doxyl group, a cyclic nitroxide with an unpaired electron, bound to the aliphatic chain carbon at position five. Paramagnetic probes induce a broadening of the NMR signals and a decrease in resonance intensities of the neighbouring protons; thus, if the protein is close to the surface or penetrates the inner core of 5-DSA spin-labelled micelles, a decrease in intensities is observed [41]. Specifically, signals of protein residues that are closer to the NO moiety are more affected by unpaired electrons.

The 2D ^1^H-^15^N-HSQC spectra of (^15^N-^13^C)-labelled ^737-786^gp36 CHR–MPER were recorded in DPC/SDS 90:10 M/M micelle solutions in the absence and presence of 5-DSA. ^1^H-^15^N-HSQC spectra of ^15^N labelled ^737-786^gp36 CHR–MPER show changes in the intensities of HN backbone signals. Figure 6 reports the variations in the backbone NH intensities (Δ%) when 5-DSA is added to the micelles. NH signals of N^756^, Q^759^, L^767^, W^776^, G^778^, I^780^, and εNH of W^770^ and W^776^ were subject to a decrease in intensity from 50% to almost 100%, indicating close contact with the lipid environment. The other residues, not being affected by any chemical shift variation, were assumed to be far from the membrane surface (Figure 6).

### 2.4. ESR Spin-Label Analysis

To determine whether the interaction with ^737-786^gp36 CHR–MPER influenced the microstructure of DPC/SDS 90:10 micelles, we performed ESR measurements on micelle solutions in the presence and absence of the peptide. Since its ESR spectrum is strongly affected by the microenvironment in which it is embedded, 5-DSA is a spin-probe suitable for this kind of study. Spectroscopic parameters obtained from the 5-DSA spectra are shown in Table 2. Two parameters can be obtained from the spectra analysis: the nitrogen isotropic hyperfine coupling constant, A_N_, which increases with the polarity of the medium in which the nitroxide is embedded, and the nitroxide correlation time, τ_C_, which depends on the label rotational mobility that is determined by the microenvironment viscosity and/or by specific interactions. Inspection of Table 1 shows that τ_C_ increases in the presence of ^737-786^gp36 CHR–MPER, indicating a reduction in the label mobility. This evidence should be ascribed to at least a partial insertion of the peptide, or of a segment of it, in the micelles, since the local structuring of the alkyl chain tends to be more ordered in the proximity of a guest molecule [55]. The local polarity, as monitored by A_N_, does not change, indicating that the peptide interaction does not favour water penetration in the micelle.

The interaction of ^737-786^gp36 CHR–MPER with the mixed DOPC/DOPG 90:10 M/M bilayer was also investigated through ESR experiments to analyse the spectra of the 5-phopshatidylcholine spin-labelled lipid (5-PCSL) included in the membrane. In this case, to quantitatively detect the perturbations of the lipid packing due to the peptide interaction, the outer hyperfine splitting, 2A_max_, which is an index of restriction of the local mobility of the acyl chains, was derived from the spectra analysis [25,31]. The 2A_max_ value of 5-PCSL increased by 1.2 G in the presence of ^737-786^gp36 CHR–MPER, revealing the stiffening of the bilayer due to the peptide interaction. A comparison between the ESR results obtained for the micelles and the liposomes shows that, in both kinds of aggregates, the interaction with the peptide significantly perturbs the surfactant (or lipid) organization, inducing a lower fluidity of the self-assembled molecules. As the reporter nitroxide group was being positioned along the tail in close proximity to the hydrophilic head group, the results indicate that the peptide penetrates at least in the region of the aggregate (micelle or bilayer) that is just underneath its interface with the aqueous medium, suggesting the involvement of the peptide in the aggregate microstructure.

### 2.5. Confocal Microscopy Imaging

Confocal microscopy imaging of ^737-786^gp36 CHR–MPER was carried out in labelled MLVs of DOPC/DOPG 90:10 M/M. DOPC included 2.25 mM of 1-palmitoyl-2-{6-[(7-nitro-2-1,3-benzoxadiazol-4-yl)amino]hexanoyl}-sn-glycero-3-phosphocholine (NBD-PC) for fluorescence visualization. Labelled MLVs were imaged in the absence and presence of ^737-786^gp36 CHR–MPER (Figure 7). Experiments were repeated at different ^737-786^gp36 CHR–MPER concentrations and at different times after deposition. Specifically, the concentrations of 0.5, 025, and 0.05 mM ^737-786^gp36 CHR–MPER were used, and images were acquired 5 min and 60 min after deposition. The best experimental conditions were obtained at 0.05 M ^737-786^gp36 CHR–MPER concentration. Figure 7A reports confocal microscopy images of MLVs composed of DOPC/DOPG 90:10 M/M in the absence of ^737-786^gp36 CHR–MPER. In these conditions, labelled MLVs assume a spherical shape, with a diameter ranging from ~1 to 10 µm.

Figure 7B show fluorescently labelled MLVs composed of DOPC/DOPG 90:10 M/M and NBD-PC in the presence of 0.05 mM ^737-786^gp36 CHR–MPER. In these conditions, the diameter of vesicles increased due to highly dynamic budding and fusion events. Large spherical liposomes characterized by 5–30 µm diameters often appear because of the impaired budding of smaller vesicles. Remarkably, labelled MLVs were interconnected through the formation of straight-chain tubular structures (Figure 7B–D) that often extend throughout the entire field of view and beyond and are typically capped at both ends with large liposomal structures. These “tubulation” events occur in different z-planes, forming a crowded three-dimensional network of tubules and liposomes.

## 3. Discussion

The MPER region of a *lentivirus* envelope glycoprotein is a hydrophobic, Trp-rich region (Figure 1), exhibiting a strong membrane affinity and an active role in the fusion of the virus envelope with the host cell membrane [1,2,3,4]. Given the critical biological role, MPER domains of different lentiviruses have been widely investigated [21,22,23,24,56,57,58,59,60]; structural data are available for gp41 MPER, and the structure of the Ebola virus envelope protein MPER/transmembrane domain (TM) has been recently determined [61]. However, notwithstanding the amount of data [24,56,57,58,59,60], many aspects of the MPER structure remain unclear, perhaps due to its conformational plasticity and chameleon-like structure.

We previously studied several peptides belonging to the MPER of FIV gp36. The 20-mer gp36 L^767-^M^786^, the octapeptide gp36 W^770^-I^777^ (C8), and the hexapeptide D^772^-I^777^(C6a) exhibited antiviral activity and were analysed using several physicochemical techniques, including NMR spectroscopy. Extending this work, we report the NMR structure determination of a small protein, L^737^-M^786^, which includes the entire gp36 MPER and part of its adjacent CHR region.

Our study provides additional data to interpret the structure-activity relationship of MPER in lentivirus glycoproteins. As structural data on gp36 are almost missing, we provide the first high-resolution structure of such an extended domain of gp36. The study of the FIV envelope glycoprotein is of great interest as it provides an experimental model to investigate HIV entry and possibly design antiviral entry inhibitors. Moreover, these data are of great interest in veterinary medicine, given the wide spread of FIV infection.

As shown in Figure 4, the structure of the ^737-786^gp36 CHR–MPER in DPC/SDS 90:10 micelles consists of a helix–turn–helix motif, where CHR and MPER are an α-helix and a 3_10_ helix, respectively. As evident from the NMR structure bundle and according to the relaxation data (Figure 5), the α-helix corresponding to part of the CHR (residues 738-757) is rigid and regular compared to the helix corresponding to MPER; the MPER helix is moderately flexible and includes residues with fast internal motion. A flexible loop (residues 758-763) connects the two helices, as demonstrated by low heteronuclear NOE values and a relatively limited number of experimental NMR restraints. However, consistent with the T_1_/T_2_ values, the two helices are oriented at an average angle of ~43°. By analysing the structural features of ^737-786^gp36 CHR–MPER in light of a structure–function relationship, it is evident that the structure of each segment fits with the relative biological function: (i) the regular CHR α-helix has a close interaction with the NHR segment (see Figure 1), (ii) the moderately flexible MPER has a less specific interaction with the lipid membrane, and (iii) the flexible CHR–MPER loop facilitates the repositioning of CHR and MPER to interact with their respective targets. The comparison of the gp36 MPER structure with the structure of the corresponding region of gp41 and Ebola envelope glycoprotein indicates that moderate flexibility is typical of all the MPER domains that have been solved thus far.

Analysis of the positioning of ^737-786^gp36 CHR–MPER on lipid membranes by using spin label-enhanced experiments indicates that the C-terminus of gp36 MPER, including W^773^ and W^776^, together with the residues flanking the loop, G^760,^ and Q^765^, lies in the membrane interface, which is well embedded in the lipid headgroup region. ESR experiments, in particular, confirm the insertion of the peptide in the micelles, as well as in the region of the DOPC/DOPG bilayer just underneath its interface with the aqueous medium (Table 1). As demonstrated by confocal microscopy imaging, this interaction destabilizes the bilayer aggregate, determining the change in size and shape of the vesicles. Figure 7 shows that ^737-786^gp36 CHR–MPER induces the highly dynamic budding and fusion events of the membrane to form a network of MLVs interconnected by straight membrane tubes.

W^773^ and W^776^ are confirmed to be important residues for effective interaction with the membrane surface. As mentioned, we previously studied the fragments of ^737-786^CHR–MPER C8, C6a, and C6b in the context of the identification of new antiviral compounds [35]. C8 and C6a correspond to W^770^-I^777^ and D^772^-I^777^ of gp36 MPER, respectively [15,19,20,27,28,29,30,31,32,33]. Both of them, including W^773^ and W^776^, exhibited antiviral activity that prevents the entry of FIV into the host cell; correspondingly, both peptides exhibited strong membrane-binding properties. On the other hand, C6b, corresponding to the W^770^-G^775^ fragment, although very similar to C8 and C6a, was almost inactive as an antiviral peptide and devoid of any relevant ability of binding to the phospholipid bilayer.

By analysing the molecular surfaces calculated as propensities of the different portions of the molecules to interact, according to the specific surface properties, we speculate that the effective interaction with a lipid membrane significantly depends on the profile of the molecular surface. While ^737-786^gp36 CHR–MPER, C8, C6a, showing a strong membrane binding property, exhibit an extended hydrophobic molecular surface derived from the exposure of the W^773^ and W^776^ indolyl rings, C6b shows a modest membrane binding property due to the limited implication of the W^770^, and W^773^ indolyl rings in interacting with the external medium. A similar analysis, applied to the MPER structure of gp41, reveals the great potential of this region to bind the lipid membrane surface due to the presence of two additional Trp residues (Figure 8).

Molecular dynamics performed on the C8 peptide using enhanced sampling approaches demonstrated that the binding of C8 with membrane phospholipids involves H-bonds and electron transfer with Trp residues and ionic interactions with the negative charges of D^772^. Using these data and those from the NMR analyses of C6a and C6b in micelle solution, we previously defined a pharmacophore model [35], including the indole rings of two Trp residues and the D^772^ carboxylate (Figure 9). The MPER structure of gp36 and gp41, as derived from NMR data, fit with this model: (i) by considering the distances among D^772^, W^770^, and W^773^ on the one hand or among D^772^, W^773^ and W^776^ on the other hand in ^737-786^gp36 CHR–MPER (Figure 9) and (ii) by considering the distances among N^674^, W^672^, and W^678^ on the one hand or N^674^, W^670^ and W^680^ on the other hand in gp41 MPER. Remarkably, gp41, due to the presence of two additional Trp residues, fits this model in more than one manner.

Taken together, these data show that MPER domains in different envelope glycoproteins and, in particular, the Trp-rich sequence within MPER may be considered polyfunctional molecular tools with amplified membrane-binding properties. These are designed by nature to manipulate phospholipid aggregates, regulating cell membrane fusion or cell membrane tubulation in important events related to cell trafficking and cell communication.

## 4. Materials and Methods

### 4.1. Protein Expression and Purification

The pET-31b(+) vector (Novagen, Cat. No. 69952-3) is designed for cloning and high-level expression of peptide sequences and is fused with the 125 aa ketosteroid isomerase protein (KSI) [62]. The construct was designed with three mutated amino acids, N737L, M751L, and G786M, to selectively remove the KSI-tag and His-tag using CNBr. The expression vector pET-31b(+) was transformed into BL21(DE3)pLysS cells and used for expression in *E. coli.* For the expression of uniformly, ^15^N and ^13^C-isotope-labelled ^737-786^gp36 CHR–MPER protein in *E. coli*, M9 minimal media [63] (6 g Na_2_HPO_4_, 3 g of KH_2_PO_4_, 0.5 g of NaCl dissolved in 900 mL of distilled water) containing 50 µg/mL ampicillin and 1 g of ^15^N ammonium sulfate was prepared and autoclaved. Then, 20% ^13^C glucose solution, prepared with 2 g of ^13^C glucose, 10 mL of vitamin cocktail and trace metal solutions (1 mL of 1 M stock solutions of thiamine, MgSO_4_, and CaCl_2_, sterilized by filtration) was added, and the final volume was adjusted to 1 L. The overnight culture (20 mL of LB (Luria Bertani) medium was used to inoculate 1 L of M9 medium. When bacterial clones reached an OD_600_ of 0.5 at 37 °C, 1 mM IPTG was added. After 18 h, cells were pelleted by centrifugation and resuspended in lysis buffer (0.5 M NaCl and 20 mM Tris-HCl), sonicated, and recentrifuged. The obtained pellet was suspended in binding buffer (10 mM guanidine, 0.5 M NaCl, 20 mM Tris-HCl, and 15 mM imidazole) and stirred for one night at 4 °C. Then, the solution was purified with a His-Trap™ HP column at 1 mL/min using an AKTA purifier system; the protein was eluted from the column with elution buffer (10 mM guanidine, 0.5 M NaCl, 20 mM Tris-HCl and 50 mM imidazole). The eluted fraction was dialyzed and lyophilized. To remove the KSI-tag [62,64], the lyophilized protein was dissolved in a minimum quantity of 70% (v/v) formic acid and treated with 0.5 M cyanogen bromide; the reaction was stirred for 3 h in the dark. After 3 h, the solution was dialyzed and lyophilized. Finally, the protein was purified by HPLC (Appendix A).

### 4.2. Mass Spectrometry Analysis

One microgram of ^737-786^gp36 CHR–MPER was digested by using 13 ng/µL of sequencing grade modified trypsin solution (Promega, USA) in 10 mM AMBIC (NH_4_HCO_3_) at 37 °C for 18 h. The reaction was stopped by adding 1 µL of 1% trifluoroacetic acid (TFA). The peptide mixture resulting from proteolytic digestion was mixed with a 1 µL matrix and spotted on a 100-well stainless-steel sample plate. The matrix was 5 mg/mL α-cyano-4-hydroxycinnamic acid dissolved in 50% acetonitrile solution (Sigma-Aldrich) with 0.1% TFA (Sigma-Aldrich). The analyses were performed on a MALDI-TOF mass spectrometer (MALDI micro MX, Waters) in the positive reflector mode. After acquisition, the spectra were externally calibrated with the calibration mixture, MassPREP™ peptide standard mixture (Waters). The analysis of the monoisotopic peptide mass lists fully covered the ^737-786^gp36 CHR–MPER sequence.

### 4.3. Circular Dichroism Spectroscopy

CD experiments were performed at 25 °C on an J-810 Jasco spectropolarimeter as an average of four scans with 10 nm/min scan speed, 4 s response time and 2 nm bandwidth, using a quartz cuvette with a path length of 1 mm, a measurement range from 190 to 260 nm (far UV), and a temperature of 25 °C. Throughout the measurements, the trace of the high-tension voltage was verified to be less than 700 V, which ensures the reliability of the obtained data.

Far UV CD spectra of mixed micelles (DPC/SDS 90:10 M/M (27 mM/3 mM)) in aqueous solution (pH 7.4, 10 mM phosphate buffer containing H_2_O), MLVs of DOPC/DOPG (90:10 M/M) and 500 µM ^737-786^gp36 CHR–MPER were added to mimic membrane systems for the conformational analysis [41]. The processed curve of ^737-786^gp36 CHR–MPER in mixed micelles was obtained by using the Spectra Analysis tool in the Jasco software. The CD curve of ^737-786^gp36 CHR–MPER in mixed micelles was corrected for the solvent contribution by subtracting the CD reference spectrum, and then the final CD spectrum was obtained after baseline correction and binomial smoothing. This resulting spectrum was used for the estimation of the secondary structure content using the algorithms CONTIN and SELCON3 from the DICHROWEB website [42].

### 4.4. NMR Spectroscopy

#### 4.4.1. Spectra Acquisition

Three-dimensional NMR experiments of ^737-786^gp36 CHR–MPER in DPC/SDS mixed micelles were acquired on a Bruker 900 MHz spectrometer. Then, 0.5 mg of isotope-labelled ^737-786^gp36 CHR–MPER was dissolved into a DPC/SDS micelle solution (DPC concentration was 27 mM (27 times higher than the critical micellar concentration (c.m.c.) of DPC), and the SDS concentration was 80 mM (10 times higher than the c.m.c. of SDS)) [65]. The DPC/SDS molar ratio was 90:10 (27 mM/3 mM) to produce a partial (2%–3%) negative charge, which is present in the typical membrane of eukaryotic cells [43]. The final pH was 7.4. For NMR experiments, ^d25^SDS and ^d38^DPC were used from Avanti^®^ Polar Lipids, Inc.

NMR experiments were recorded at 300 K. Standard backbone, and side-chain assignment experiments (CBCA(CO)NH, HNCACB, HNCO, HN(CA)CO, HBHA(CO)NH, and HCCH-TOCSY) [46] were acquired with 1024 increments of 512 time points and 16 scans each on a 600 MHz Bruker Avance III spectrometer equipped with a TXI probe, running with TopSpin 2.1. On this spectrometer, T_1_, T_2_, and relaxation experiments were also recorded. T_1,_ relaxation curves were fit from signal intensities in HSQC experiments recorded as pseudo-3D experiments using relaxation delays of 10, 20, 40, 80, 160, 400, 1000, 2000, 4000, 6000 and 7000 ms. T_2_ times were obtained from fitting HSQC signal intensities after recording pseudo-3D experiments with 31.7 (2 times), 63.4, 95.1, 126.8 (2 times), 158.6, 190.3, 222.0, 253.7 and 285.4 ms. The 2D NOESY, 3D ^13^C-NOESY-HSQC, and ^15^N-NOESY-HSQC experiments were recorded with 100 ms mixing times using a 900 MHz Bruker Avance III NMR spectrometer equipped with a TCI cryoprobe and running with TopSpin 3.0.

#### 4.4.2. Assignment of NMR Resonances and TALOS+ Analysis

Qualitative and quantitative analyses of 3D NMR spectra were achieved using SPARKY software [44]. Intramolecular distance restraints derived from nuclear Overhauser enhancements (NOEs) were obtained from the ^15^N-NOESY spectrum recorded on a 900 MHz spectrometer. The assigned chemical shift values of backbone ^15^N, ^13^Cα, and ^13^C’ were used as input for the TALOS+ program [50] to predict backbone dihedral angles. Forty-two out of 44 residues with assignments and dihedral angles of 42 residues were considered ‘GOOD’ by TALOS+.

#### 4.4.3. Structure Calculation

Peak volumes were translated into upper distance bounds with the CALIBA routine from the CYANA 2.1 software package [51]. After discarding redundant and duplicated constraints, the final list of constraints was used to generate an ensemble of 50 structures by the standard CYANA protocol of simulated annealing in torsion angle space (using 6000 steps). The entries that present the lowest target function value (0.83–1.19) and small residual violations (maximum violation = 0.38 Å) were analysed using the PyMOL program [66]. 

#### 4.4.4. NMR Spin-Label Experiments

2D ^1^H-^15^N-HSQC experiments of 0.5 mg of ^737-786^gp36 CHR–MPER in DPC/SDS mixed micelles were acquired in the absence and in the presence of 5-doxylstearic acid (5-DSA). 5-DSA purchased from Merck was withdrawn from a 10 mM methanol-d_4_ stock solution. All experiments were acquired with 512 increments of 256 time points and 32 scans on a 600 MHz Bruker Avance III spectrometer equipped with a TXI probe, running with TopSpin 2.1.

### 4.5. ESR Spectroscopy

The effect of ^737-786^gp36 CHR–MPER on the surfactant arrangement in DPC/SDS 90:10 micellar aggregates was investigated by analysing the ESR spectrum of the spin label 5-doxylstearic acid (5-DSA; Sigma). Moreover, the effect of ^737-786^gp36 CHR–MPER on lipid self-organization in DOPC/DOPG 90:10 liposomes was investigated by analysing the ESR spectrum of the spin label 1-palmitoyl-2-stearoyl-(5-doxyl)-sn-glycero-3-phosphocholines (5-PCSL; Avanti Polar Lipids). In both micellar and liposomal systems, the peptide/surfactant (peptide/lipid) molar ratio was set at 1:50, and the spin label was inserted into the self-assembled aggregates at a 1:100 label/surfactant (label/lipid) molar ratio, with a fixed total surfactant (or lipid) concentration of 30 mM.

The samples, prepared according to well-assessed procedures reported in previous works [67,68], were deoxygenated and sealed in 1.00 mm i.d. quartz capillaries. ESR spectra were obtained using a Bruker ELEXYS e500 X-band spectrometer using instrumental settings reported elsewhere [55,69]. All measurements were performed at 25 ± 1 °C.

ESR spectra of 5-DSA in micellar systems showed narrow lines indicative of an isotropic fast motion. Consequently, the spectral analysis was carried out according to the classical theory of motional narrowing for ESR lines using a line-shape analysis protocol reported in the literature. On the other hand, the ESR spectra of 5-PCSL in liposomes showed a slightly, but evidently anisotropic, line shape. In this case, semiquantitative analysis of the spectra was carried out by determining the outer hyperfine splitting, *2A_max_*, which is the difference between the low-field maximum and the high-field minimum [25].

### 4.6. Confocal Microscope Imaging

The multilamellar lipid vesicles (MLVs) of DOPC/DOPG (90:10, M/M) with 2.25 mM NBD-PC were prepared by mixing appropriate amounts of lipids dissolved in a dichloromethane/methanol mixture (25 mM lipid concentration) in a round-bottom test tube. The solutions were desiccated, dried overnight, and hydrated in 10 mM phosphate buffer at pH 7.4. The total weight of the lipid for each sample was 1.0 mg. During their formation, labelled MLVs were prepared by adding ^737-786^gp36 CHR–MPER at concentrations of 0.50, 0.25 and 0.05 mM. The last concentration, corresponding to the lipid-to-peptide ratio of 50:0.1 M/M, was used for the imaging shown in Figure 7.

A total of 10 µL of each solution, in the presence or absence of ^737-786^gp36 CHR–MPER, was taken and spotted onto a cover slip. Images were acquired as previously described [70,71] at 5 and 60 min after deposition on a laser scanning confocal microscope (LSM 510; Carl Zeiss Micro-Imaging) equipped with a plan Apo 63X, NA 1.4 oil immersion objective lens. For each field, both fluorescent and transmitted light images were acquired on separate photomultipliers and were analysed using Zeiss LSM 510 4.0 SP2 software. In samples in which different z-planes were distinguishable, a z-stack acquisition mode was performed to focus a single z-plane, as published previously [72].

## Figures and Tables

**Figure 1 ijms-21-02037-f001:**
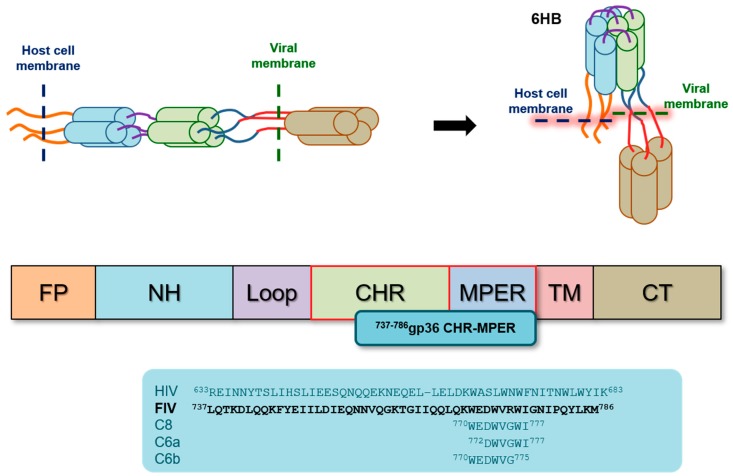
The structural framework and conformational rearrangement of gp41 and gp36 from a pre-hairpin state to the six-helical bundle (6HB). Amino acid sequences of HIV gp41 C-terminal heptad repeat–Trp-rich membrane-proximal external region (CHR–MPER), feline immundeficiency virus (FIV) ^737-786^gp36 CHR–MPER, and FIV-derived peptides.

**Figure 2 ijms-21-02037-f002:**
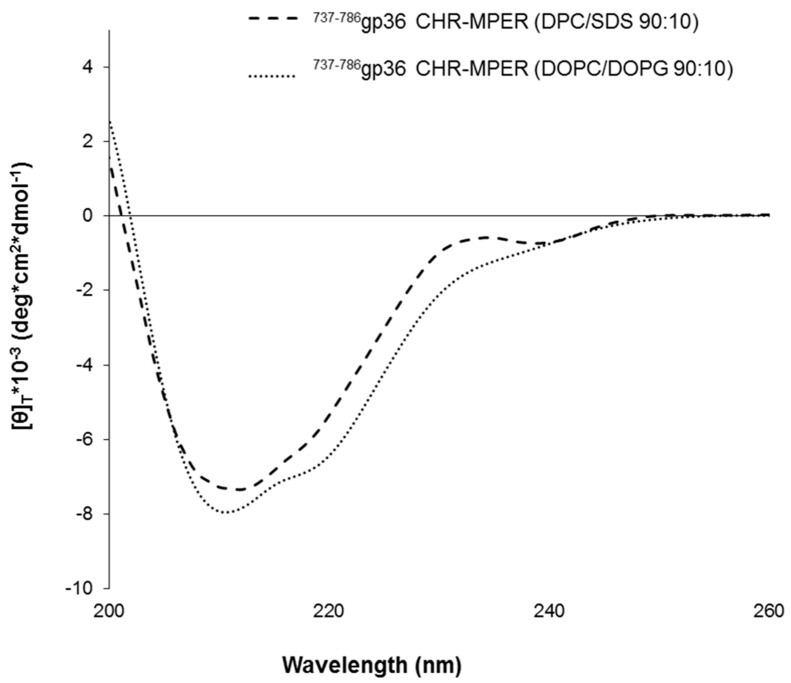
Processed far-UV CD spectra of ^737-786^gp36 CHR–MPER in multilamellar vesicles (MLVs) composed of dodecyl phosphocholine/sodium dodecyl sulfate (DPC/SDS) 90:10 M/M (dashed line) or 1,2-dioleoyl-sn-glycero-3-phosphocholine/1,2-dioleoyl-sn-glycero-3-phospho-(1’-rac-glycerol) (DOPC/DOPG) 90:10 M/M (dotted line). CD spectra were acquired using a JASCO-810 spectropolarimeter at room temperature with a cell path length of 1 mm. The measurement range spans from 200 to 260 nm.

**Figure 3 ijms-21-02037-f003:**
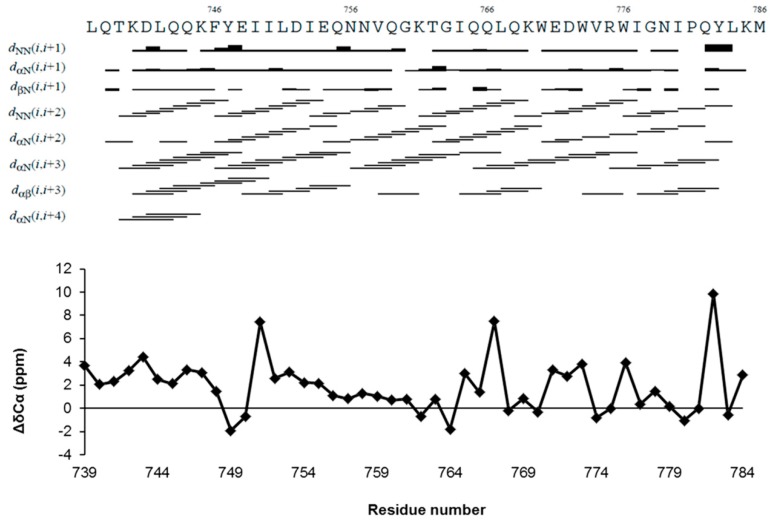
Top: an overview of the sequential and medium-range nuclear Overhauser enhancements (NOEs) used to calculate the ^737-786^gp36 CHR–MPER structure ensemble. Bottom: ^13^Cα secondary chemical shifts (ΔδCα) of ^737-786^gp36 CHR–MPER in 100 mM DPC/SDS (90:10 M/M).

**Figure 4 ijms-21-02037-f004:**
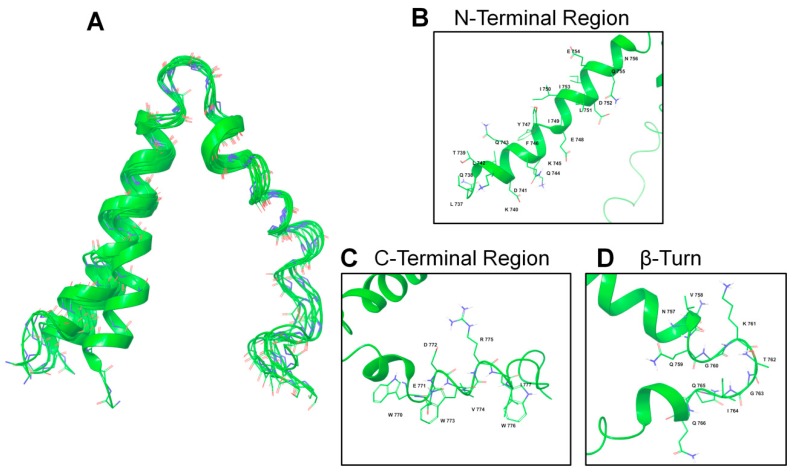
NMR structure of (^15^N-^13^C)-labelled ^737-786^gp36 CHR–MPER in DPC/SDS 90:10 M/M. (**A**) Ribbon of the representative structure of the calculated ensemble; (**B**) N-terminal region, consisting of the N-helix residues Q^738^-N^756^; (**C**) C-terminal region containing the C8 peptide; (**D**) β-turn region containing residues N^757^–Q^766^.

**Figure 5 ijms-21-02037-f005:**
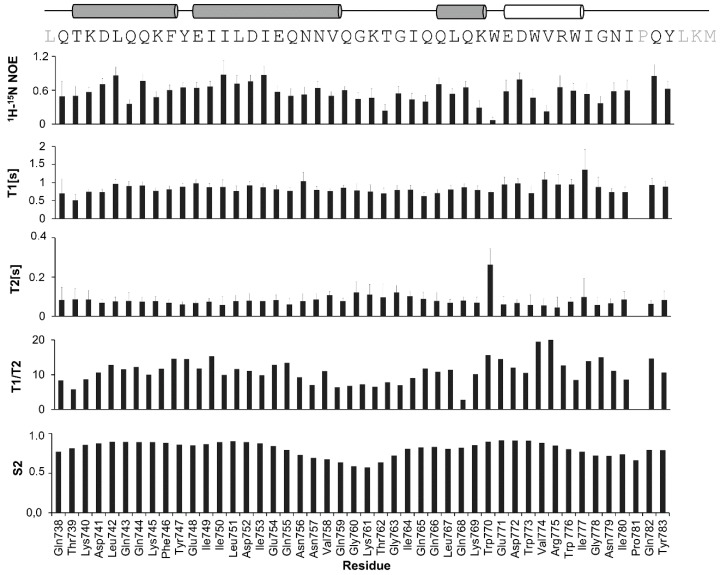
^15^N backbone relaxation measurements for ^737-786^gp36 CHR–MPER in DPC/SDS micelles. ^1^H-^15^N heteronuclear NOE, T_1,_ and T_2_ experiments were performed on a 600 MHz spectrometer (see Materials and Methods for experimental details). Plot of the T_1_/T_2_ ratios obtained from T_1_ and T_2_ data. Predicted random coil index (RCI) histogram plotted from S_2_ values obtained using the TALOS+ routine (see Appendix A for data in detail). The α-helices are reported as grey cylinders in the one-letter sequence of ^737-786^gp36 CHR–MPER at the top, while the 3_10_ helix is reported as a white tube.

**Figure 6 ijms-21-02037-f006:**
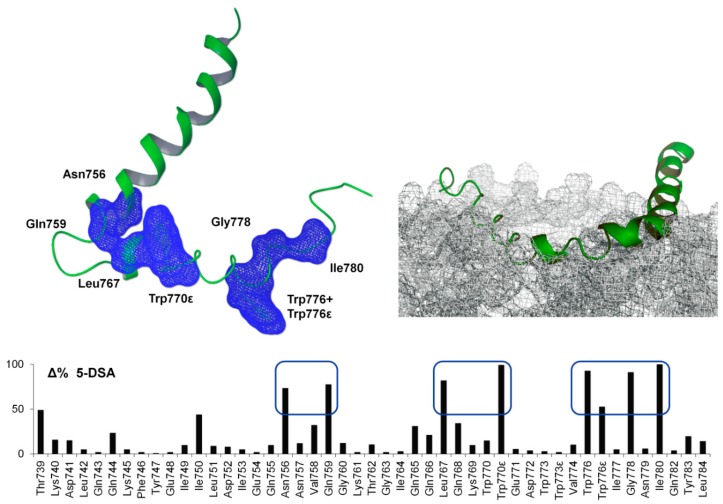
Percentage variation (Δ%) of intensities of HN backbone peaks of (^15^N-^13^C)-labelled ^737-786^gp36 CHR–MPER in the DPC/SDS 90:10 M/M micelle solution obtained by integrating 2D ^1^H-^15^N-HSQC spectra recorded at 300 K in the absence and presence of 5-doxylstearic acid (5-DSA) at a concentration of one spin label per micelle. The residues that show a decrease in intensity from 50 to 100% are highlighted in blue. At the top right is a model of the interaction between ^737-786^gp36 CHR–MPER and micelles.

**Figure 7 ijms-21-02037-f007:**
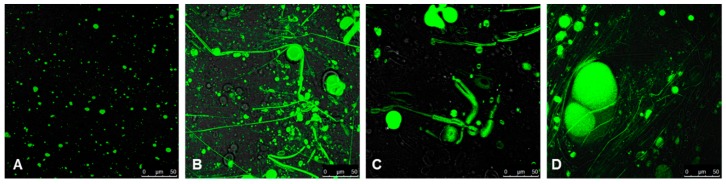
Confocal microscopy images of DOPC/DOPG 90:10 M/M labelled MLVs (**A**) and DOPC/DOPG 90:10 M/M labeled MLVs in the presence of ^737-786^gp36 CHR–MPER (**B**–**D**). In B–D, the same sample is imaged in different regions of the field. DOPC phospholipids include 2.25 mM NBD-PC. Images were acquired on a laser scanning confocal microscope (LSM 510; Carl Zeiss Micro-Imaging) equipped with a plan Apo 63×, NA 1.4 oil immersion objective lens. For each field, both fluorescent and transmitted light images were acquired on separate photomultipliers and were analysed using Zeiss LSM 510 4.0 SP2 software.

**Figure 8 ijms-21-02037-f008:**
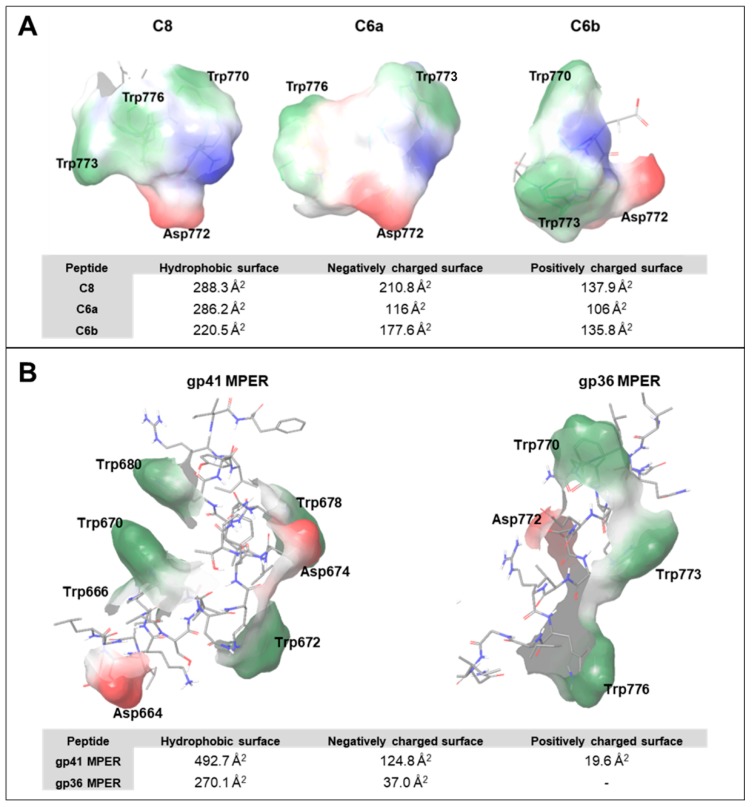
Surface analyses of the Trp and Asp residues contained in (**A**) C8, C6a, and C6b peptides and (**B**) gp41 and gp36 MPER domains. The patches are coloured green for hydrophobic surfaces, red for negatively charged surfaces, and blue for positively charged surfaces.

**Figure 9 ijms-21-02037-f009:**
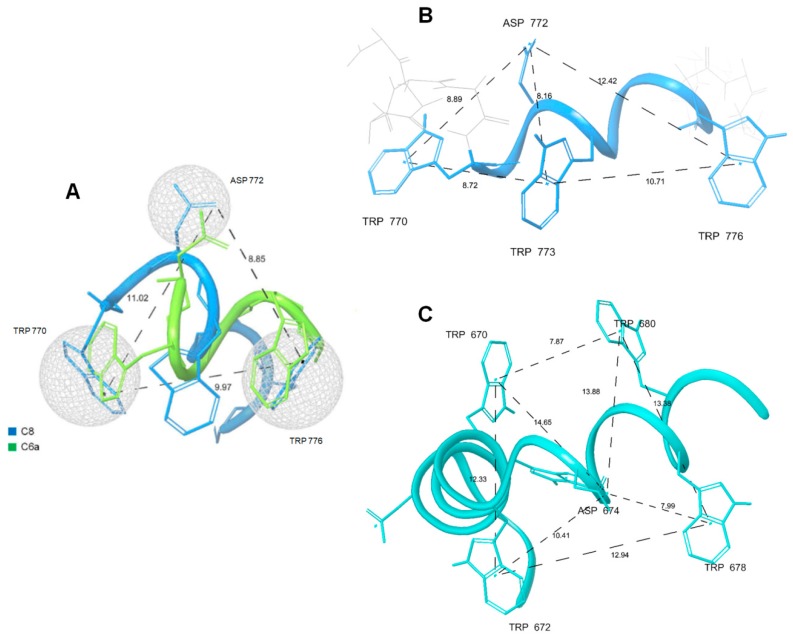
The structures of gp36 MPER (**B**) and gp41 MPER (**C**) fit the pharmacophore model of C8 and C6a previously identified [35] (**A**). Dashed lines show the distances in Å between the key functional moieties.

**Table 1 ijms-21-02037-t001:** Statistics for the structural calculation of the NMR Ensemble of ^737-786^gp36 CHR–MPER.

**Number of NOESY Peaks**	
Total	457
**Number of Experimental Restraints after CYANA**	
Total NOEs	445
Intra-residual	294
Sequential	122
Medium-range	20
Long-range	9
Dihedral angles	73
***RMSD***	
bb/heavy Å	1.38/1.98
**Ramachandran Analysis**	
Favourable regions	78.40%
Additionally allowed regions	19.30%
Generously allowed regions	2.30%
Disallowed regions	0%

**Table 2 ijms-21-02037-t002:** Nitrogen coupling constant, *A*_N_, spin-label correlation time, τ_C_, for 5-DSA in micelles and outer hyperfine splitting, 2*A*_max_, for 5-PCSL measured by ESR experiments in the absence and presence of ^737-786^gp36 CHR–MPER.

	*A*_N_/G	τ_C_ × 10^10^/s	2*A*_max_/G
***5-DSA***			
DPC/SDS 90:10	14.5 ± 0.1	27 ± 2	-
DPC/SDS 90:10 + ^737-786^gp36 CHR–MPER	14.6 ± 0.1	42 ± 3	-
***5-PCSL***			
DOPC/DOPG 90:10	-	-	51.2 ± 0.1
DOPC/DOPG 90:10 + ^737-786^gp36 CHR–MPER	-	-	52.4 ± 0.2

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
