# Peer review of "NMR Structure of the FIV gp36 C-terminal Heptad Repeat and Membrane-Proximal External Region"

_ijms, 2020, doi:10.3390/ijms21062037_

Round 1

Reviewer 1 Report

The authors have addressed all of my questions.

Author Response

Response to Reviewer 2 Comments

Point 1: The number of scans and number of increments for the NMR spectra should be included in the materials and methods.

Point 2: Can you include experimental details about the PRE experiments in the materials and methods.

 Response 1-2: We modified materials and methods including the experimental details requested from the reviewer.

In figure 6, the PRE effects on the amide peak intensities are recorded but several residues seem to have no PRE effect, this seems unlikey as there will usually be a slight effect for the residues neighbouring the membrane interacting residues. I assume that there some spectral quality issues or overlap that preclude the analysis of these residues, could you comment on this in the figure legend or indicate any unobservable/overlapped residues in the figure itself.

Response 2: We repeated the analysis of HSQC spectra. Indeed there are not spectral quality issues or overlap, as evident from the image of the spectra that we report below. However by repeating spectral analysis, we found the incorrect evaluation of peak intensity for those residues that seem to have no PRE effect. In these cases, we observed modest but observable changes in peak intensity. The graph in figure 6 has been replaced consistently with the new analysis.

Reviewer 2 Report

The paper is much improved by the edits that have been made, I have a couple of further comments:

The number of scans and number of increments for the NMR spectra should be included in the materials and methods.

Can you include experimental details about the PRE experiments in the materials and methods. In figure 6, the PRE effects on the amide peak intensities are recorded but several residues seem to have no PRE effect, this seems unlikey as there will usually be a slight effect for the residues neighbouring the membrane interacting residues. I assume that there some spectral quality issues or overlap that preclude the analysis of these residues, could you comment on this in the figure legend or indicate any unobservable/overlapped residues in the figure itself. 

Author Response

(The authors gave the same response as above.)
